# Relativistic and resonant effects in the ionization of heavy atoms by ultra-intense hard X-rays

Benedikt Rudek[1], Koudai Toyota[2], Lutz Foucar[3], Benjamin Erk [4], Rebecca Boll [5,6], Cédric Bomme[4], Jonathan Correa[2,4], Sebastian Carron[7,8], Sébastien Boutet[7], Garth J. Williams[7,9], Ken R. Ferguson[7], Roberto Alonso-Mori [7], Jason E. Koglin [7], Tais Gorkhover[7,10], Maximilian Bucher [7,11], Carl Stefan Lehmann[11,12], Bertold Krässig[11], Stephen H. Southworth[11], Linda Young[11,13], Christoph Bostedt[11,14], Kiyoshi Ueda[15], Tatiana Marchenko[16], Marc Simon[16], Zoltan Jurek[2], Robin Santra [2,17], Artem Rudenko[18], Sang-Kil Son [2] & Daniel Rolles[4,18]

An accurate description of the interaction of intense hard X-ray pulses with heavy atoms, which is crucial for many applications of free-electron lasers, represents a hitherto unresolved challenge for theory because of the enormous number of electronic configurations and relativistic effects, which need to be taken into account. Here we report results on multiple ionization of xenon atoms by ultra-intense (about $10^{19}$ W/cm$^2$) femtosecond X-ray pulses at photon energies from 5.5 to 8.3 keV and present a theoretical model capable of reproducing the experimental data in the entire energy range. Our analysis shows that the interplay of resonant and relativistic effects results in strongly structured charge state distributions, which reflect resonant positions of relativistically shifted electronic levels of highly charged ions created during the X-ray pulse. The theoretical approach described here provides a basis for accurate modeling of radiation damage in hard X-ray imaging experiments on targets with high-$Z$ constituents.

[1] Physikalisch-Technische Bundesanstalt, Braunschweig, Germany. [2] Center for Free-Electron Laser Science, DESY, Hamburg, Germany. [3] Max Planck Institute for Medical Research, Heidelberg, Germany. [4] Deutsches Elektronen-Synchrotron (DESY), Hamburg, Germany. [5] Max Planck Institute for Nuclear Physics, Heidelberg, Germany. [6] European XFEL GmbH, Schenefeld, Germany. [7] LCLS, SLAC National Accelerator Laboratory, Menlo Park, CA, USA. [8] California Lutheran University, Thousand Oaks, CA, USA. [9] NSLS-II, Brookhaven National Laboratory, Upton, NY, USA. [10] Stanford PULSE Institute, SLAC, Menlo Park, CA, USA. [11] Argonne National Laboratory, Lemont, IL, USA. [12] Fachbereich Chemie, Philipps-Universität Marburg, Marburg, Germany. [13] Department of Physics and The James Franck Institute, University of Chicago, Chicago, IL, USA. [14] Department of Physics and Astronomy, Northwestern University, Evanston, IL, USA. [15] Institute of Multidisciplinary Research for Advanced Materials, Tohoku University, 2-1-1 Katahira, Sendai, Japan. [16] Laboratoire de Chimie Physique—Matière et Rayonnement, LCPMR, CNRS, Sorbonne Université, Paris, France. [17] Department of Physics, University of Hamburg, Hamburg, Germany. [18] J.R. Macdonald Laboratory, Department of Physics, Kansas State University, Manhattan, KS, USA. These authors contributed equally: Benedikt Rudek, Koudai Toyota. Correspondence and requests for materials should be addressed to D.R. (email: rolles@phys.ksu.edu)

The availability of ultra-intense and short-pulsed free-electron laser radiation in the hard X-ray domain has opened up numerous avenues for investigating the structure and dynamics of matter with atomic resolution[1–4]. For hard X-rays, the interaction of light with a polyatomic system is often dominated by the presence of heavy (high-Z) elements since their absorption cross-section is typically two to three orders of magnitude larger than for light atoms such as carbon, nitrogen, and oxygen[5]. Heavy atoms, which occur naturally or can be embedded artificially in many molecules of biological relevance, play a particularly important role for imaging applications because their strong contribution to the total scattering signal is commonly used for phasing of the X-ray diffraction patterns[6]. Moreover, it was suggested that anomalous-diffraction effects caused by heavy atoms can provide a clean solution to the crystallographic phase problem even at very high X-ray intensities[7,8].

When a single X-ray photon is absorbed by a heavy atom, an electron is typically emitted from one of the strongly bound inner shells. The electron vacancy is filled by an electron from the higher-lying shells, and the energy difference of this transition is dissipated radiatively (fluorescence) or via emission of yet another electron (Auger decay). This often triggers the emission of further electrons via an Auger cascade[9]. In an intense X-ray free-electron laser (XFEL) pulse, several X-ray photons can be absorbed by a single atom, leading to the creation of extremely high charge states[10–12]. Early studies have shown that multiphoton ionization of light elements by an XFEL pulse can be described theoretically by considering a sequence of single-photon transitions to the continuum[10,13]. For heavy atoms, it was observed that resonant excitations of inner-shell electrons into densely-spaced Rydberg states and unoccupied valence orbitals can dramatically enhance the ionization in the soft X-ray regime via a process dubbed resonance-enabled or resonance-enhanced X-ray multiple ionization (REXMI)[11,12]. The enhancement has been qualitatively reproduced by theoretical models taking into account bound-bound resonant transitions in the intermediate ionic states[14–16]. However, a quantitative description of the resonant enhancement, which is essential, for example, for the proposed high-intensity phasing schemes[7,8], requires a relativistic treatment[17]. This is extremely challenging since the combination of deep inner-shell ionization, relativistic effects, and resonant excitations creates an enormous computational complexity, especially at high X-ray intensities, where more than 20 high-energy photons can be absorbed by a single atom.

Here, we present experimental data on the multiple ionization of Xe atoms by ultra-intense XFEL pulses at several hard X-ray wavelengths and, by comparison with theory, demonstrate that relativistic and resonant effects indeed play a crucial role in this regime. They significantly enhance ionization and give rise to surprisingly structured charge state distributions (CSDs), which reflect resonance positions of relativistically shifted electronic levels of highly charged ions reached during the X-ray pulse as a consequence of the X-ray-driven ionization dynamics. The good agreement between experimental and theoretical results demonstrates that in spite of the complexity of the problem, our current ionization model can fully explain the experimental findings for all X-ray beam parameters.

## Results

**Experimental charge state distributions**. Our experiments (see Methods) were carried out at the hard-X-ray nanometer-focus end-station of the Linac Coherent Light Source (LCLS) with peak fluences up to 5.1 mJ/$\mu$m$^2$, which is two orders of magnitude higher than in previous measurements[11,12,18,19]. A dilute target of Xe atoms was exposed to the X-ray pulses, and the resulting

atomic CSDs were measured using ion time-of-flight (ToF) mass spectrometry.

Figure 1 depicts the ion ToF spectrum of Xe atoms ionized with hard X-ray FEL pulses at different photon energies (all of which were chosen to be above the L-shell ionization thresholds of neutral Xe) but with the same nominal pulse energy. While the maximum observed charge state increases steadily as the photon energy increases, the yield for specific charge states varies non-monotonically as a function of photon energy. The CSDs for 5.5, 6.5, and 8.3 keV photon energy are shown as black squares in Figs. 2a and 3a, b, respectively. The pulse energies and corresponding peak fluence values are given in Table 1 in the Methods section. At all three photon energies, the most abundant charge state is Xe$^{8+}$, which is characteristic for single-photon ionization of the Xe $2s$ and $2p$ levels followed by an Auger cascade[9]. This strong contribution from single-photon ionization stems from the broad low-fluence areas in the wings of the X-ray focus profile. Charge states of Xe$^{12+}$ and beyond result from multiphoton ionization, and their yield mostly decreases towards higher charge states. At 5.5 keV, three broad peaks around Xe$^{27+}$, Xe$^{31+}$, and Xe$^{37+}$ stand out from this steady decrease, as seen in Fig. 2a.

**Relativistic and resonant effects**. In order to interpret the observed CSDs and to elucidate the origin of the peak structure, we performed ab initio ionization dynamics calculations based on a rate-equation approach using the XATOM toolkit[20–22], which was recently extended to include resonant transitions and relativistic effects[17]. The former is essential to reproduce the resonant enhancement, while the latter is particularly relevant for heavy-

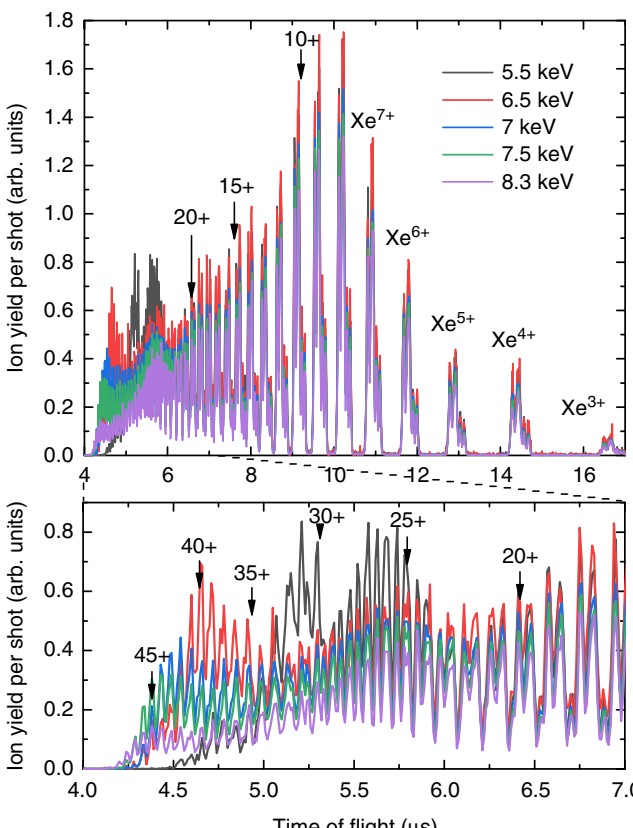

**Fig. 1** Ion time-of-flight spectra of atomic xenon. The spectra were recorded at photon energies from 5.5 to 8.3 keV at the same nominal pulse energy of 3.7 ± 0.05 mJ. The lower graph magnifies the region of high charge states. The arrows indicate the $n$-th xenon charge state, $^{132}$Xe$^{n+}$

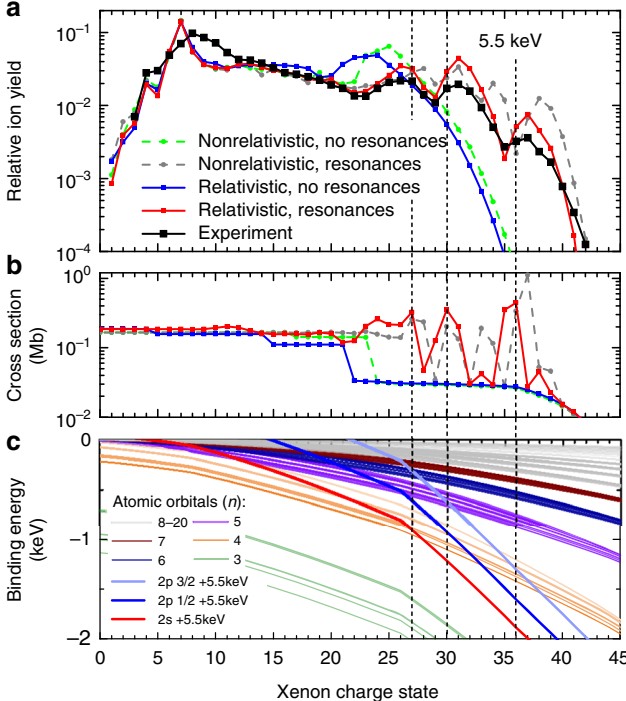

**Fig. 2** Peak structure in the Xe charge state distributions. **a** Experimental and calculated Xe charge state distributions at 5.5 keV. The calculations were carried out with and without consideration of relativistic effects and resonances. The sum of all charge states is normalized to one. The statistical uncertainties for both theory and experiment are smaller than 1% except for the experimental data starting at +40, which have a maximum uncertainty of 3% for the highest charge state, +42. The error bars are, however, smaller than the size of the symbols. **b** Calculated total photoabsorption cross-section of the ground state Xe ion at 5.5 keV and 1% bandwidth with and without relativistic effects and resonances. **c** Binding energy of the ground-state Xe orbitals including relativistic effects drawn as green, orange, purple, dark blue, and dark red lines for the $n = 3$ to $n = 7$ orbitals. Higher orbitals are drawn in gray. The red, blue, and pale blue lines are offset from the $2s$, $2p_{1/2}$, and $2p_{3/2}$ orbital energies by the 5.5-keV photon energy, respectively. The positions where these lines cross the $n = 4$ levels (orange lines) mark the positions of resonances and are indicated by dashed lines across all three panels

atom ionization, where the spin–orbit splitting of the Xe $2p$ level, for example, which is about 300 eV, opens up new decay channels such as Coster–Kronig transitions. The coupled rate equations, whose total number exceeds $10^{68}$ in the present study, are solved via a Monte Carlo on-the-fly approach[17,18] (see Methods).

In Fig. 2a, the theoretical results with and without including resonance and relativistic effects (see color coding in the legend) are presented in comparison with the experimental CSD (black line and symbols). For a quantitative comparison, the theoretical data are averaged over the fluence distribution in the focal volume (see Methods). The overall behavior and the number and position of the peaks in the experimental CSD are only reproduced by the calculations when both resonant transitions and relativistic effects are included (red). Without resonant transitions (green and blue), the ion yield rapidly drops after the charge state whose $2p$ binding energy surpasses the 5.5-keV photon energy. The non-relativistic, resonant calculation (gray) shows more than three peaks, and the peak positions are not matched with the experimental observation.

The resonant enhancements occur when the binding energy of the $L$-shell orbitals in the highly charged ions exceeds the photon energy, such that further direct ionization from the $L$-shell is no longer possible. However, an $L$-shell electron can undergo a bound–bound transition into one of the unoccupied levels, followed by Auger decay, typically leaving the excited electron as a spectator (see Supplementary Figure 2 and Supplementary Table 1). This resonant Auger process may happen multiple times and drive a REXMI mechanism[11,12]. As can be seen from comparison with theory, the resonantly enhanced multiphoton ionization drastically extends the highest achievable charge state at 5.5 keV photon energy up to $Xe^{42+}$ and, moreover, leads to three pronounced peaks around $Xe^{27+}$, $Xe^{31+}$, and $Xe^{37+}$. Both of these effects were not observed in the earlier experiment on Xe at 5.5 keV[18] because of much lower fluence values in that experiment.

To elucidate the underlying physics of the peak structure, we calculated the total photoabsorption cross-sections for the electronic ground state of each charge state, shown in Fig. 2b. For the resonant calculations, the cross-sections were convolved with a 1% bandwidth, corresponding to the approximate photon bandwidth of the experiment. The corresponding orbital binding energies are plotted in Fig. 2c for the relativistic case. The red, blue, and pale blue lines are offset from the $2s$, $2p_{1/2}$, and $2p_{3/2}$

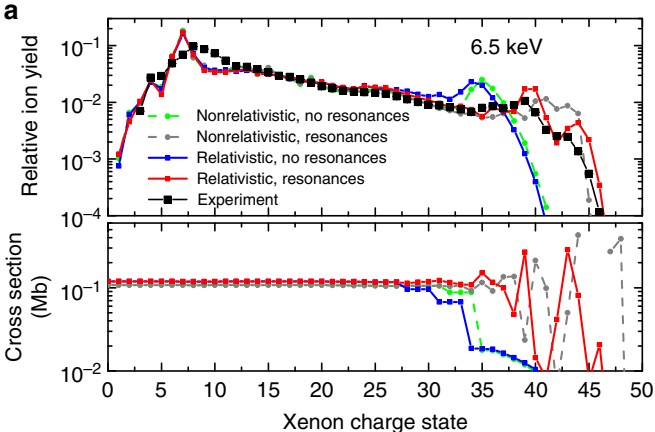

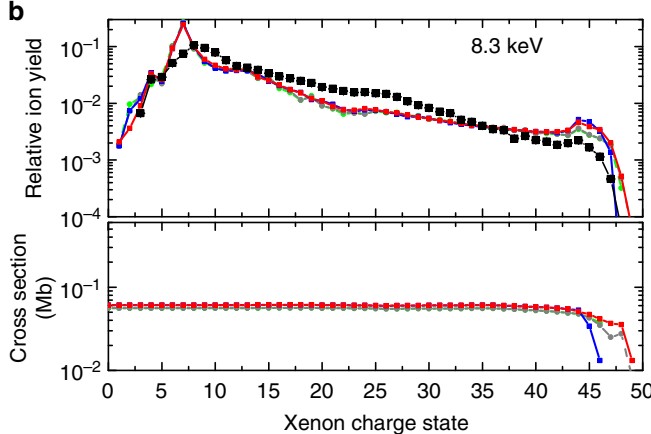

**Fig. 3** Xe charge state distributions at higher photon energies. Experimental and calculated Xe charge state distributions at **a** 6.5 keV and **b** 8.3 keV photon energy. The calculated total photoabsorption cross-sections of a ground state Xe ion for 1% bandwidth are plotted in the lower panels. The charge state distributions and photoabsorption cross-sections were calculated with and without consideration of relativistic effects and resonances. The sum of all charge states is normalized to one. The statistical uncertainties for both theory and experiment are smaller smaller than the size of the symbols

**Table 1 X-ray beam parameters**

| Photon energy (keV) | Pulse energy (mJ) | Number of shots | Peak fluence (mJ/$\mu$m$^2$) | Width ratio | Fluence ratio | Transmission (%) |
|---|---|---|---|---|---|---|
| 5.5 | 4.0–4.2 | 29,100 | 2.21 | 2.52 | 0.64 | 19.8 |
| 6.5 | 4.2–4.4 | 17,347 | 4.09 | 3.10 | 0.28 | 32.6 |
| 8.3 | 3.4–3.6 | 15,610 | 5.12 | 3.27 | 0.12 | 35.9 |

The following parameters were used for the calculations in Figs. 2 and 3. The pulse energy and the number of shots are taken from experiment. The peak fluence, the width ratio, and the fluence ratio are fitted from the calibration, and the beamline transmission is calculated with these parameters, assuming the same focus size of 0.35 μm × 0.3 μm (FWHM). No calibration data are available for the photon energies of 7 and 7.5 keV used in Fig. 1

orbital energies by the 5.5-keV photon energy, respectively. The positions where these lines cross the $n = 4$ levels (orange lines) mark the positions of resonances. Around those resonance positions, the calculated cross-sections are enhanced up to 30-fold, which give rise to resonant Auger processes and thus the REXMI effect[11]. Since the $N$-shell ($n = 4$) is fully occupied in neutral Xe, these resonant transitions to the $N$-shell of the multiply charged ions can be considered as hidden resonances[23] as they do not occur in neutral Xe. Further analysis of the resonant photoabsorption cross-sections for individual transitions (see Supplementary Figure 3) reveals that for the relativistic case, the maxima of the cross-sections are at Xe$^{27+}$ due to $2s_{1/2} \rightarrow 4p_{1/2}/4p_{3/2}$ transitions, at Xe$^{30+}$ due to $2p_{1/2} \rightarrow 4d_{3/2}$ transitions, and at Xe$^{36+}$ due to $2p_{3/2} \rightarrow 4d_{5/2}$ transitions, which are highlighted by dashed lines extending over all figure panels. As a consequence of the increased cross-section, the CSD also exhibits three maxima centered at the charge states directly following these resonance positions, except for the first maximum around Xe$^{26+}$, where several transitions from the $L$-shell to $n > 4$ also contribute at slightly lower charge states. The three distinct peaks can only be accurately reproduced by the calculations if relativistic effects such as spin–orbit splitting are included, and their experimental observation is thus a direct probe of the relativistically shifted electronic levels of highly charged ions. The non-relativistic, resonant calculation (gray) produces four peaks (mainly corresponding to $2s \rightarrow 5p$, $2p \rightarrow 5d$, $2s \rightarrow 4p$, and $2p \rightarrow 4d$ transitions) instead of three peaks and, thus, does not reproduce the experimental data. We note that in the low-charge-state region below Xe$^{13+}$, which is dominated by single-photon ionization, all four calculations show deviations from the experimental CSD as a result of neglecting higher-order many-electron corrections. Such effects are expected to lose relevance for higher charge states, where fewer and fewer electrons remain bound.

Figure 3 shows the experimental CSDs for X-ray energies of (a) 6.5 keV and (b) 8.3 keV, along with similar calculations as described above. As for 5.5 keV, photoionization predominantly proceeds via ionization of the $2s$ and $2p$ orbitals, so that the CSDs at all three photon energies look very similar up to Xe$^{22+}$. At 6.5 keV, the $2p_{3/2}$ binding energy surpasses the photon energy at Xe$^{35+}$ (non-relativistic) and Xe$^{34+}$ (relativistic), and the cross-section shows a peak structure at subsequent charge states due to resonant transitions. Again, the peak structure in the experimental data is well reproduced only by the relativistic, resonant calculation. At 8.3 keV, direct $L$-shell ionization only closes at Xe$^{47+}$, and the photon energy is sufficiently far above the $2p$ binding energy for a wide range of charge states such that there is almost no difference between the relativistic and the non-relativistic cross-sections. As a result, the theoretical CSDs from all four calculations at 8.3 keV are very similar to each other.

## Discussion

Based on the good agreement of the CSDs obtained from experiment and theory including relativistic and resonance effects, we are able to quantitatively describe and predict the degree of ionization for heavy atoms after the interaction with intense hard X-rays as a function of fluence. Figure 4a presents the fluence-dependent CSDs at 5.5 keV without focal volume averaging (the peak-fluence dependence of the experimental charge state yields is given in Supplementary Figure 4 and discussed in the Supplementary Discussion). The calculations show that the three characteristic peaks emerge at different fluence regimes, which contribute to the experimentally observed CSD due to volume averaging. Further insight is provided by Fig. 4b, which shows the mean charge calculated from the final ion yields as a function of fluence at three different photon energies. The final mean charge is an important atomic benchmark in the formation of warm or hot dense matter and in the electronic radiation damage for molecular imaging. As can be seen in Fig. 4b, the final mean charge increases as the fluence increases, but it reaches saturation at different fluences for different photon energies. At 5.5 keV (red curve), saturation starts early (around 0.5 mJ/$\mu$m$^2$) when direct $L$-shell photoionization is no longer available. Further charging-up occurs via REXMI as well as via valence-shell photoionization and multiple-core-hole state formation, and it fully saturates when no further resonances can occur and all valence electrons are depleted, which corresponds to a charge state of Xe$^{44+}$. On the other hand, direct $L$-shell photoionization at 8.3 keV closes at Xe$^{47+}$, and the 8.3-keV curve (green) therefore saturates around 5 mJ/$\mu$m$^2$, eventually approaching Xe$^{50+}$. Interestingly, the plot shows crossings of the curves in the fluence region between 1–5 mJ/$\mu$m$^2$. At lower fluences, lower photon energies induce a higher final charge because of higher single-photon ionization cross-sections. In the high fluence regime (>5 mJ/$\mu$m$^2$), however, the opposite trend is observed, and higher photon energies lead to a higher final mean charge. This counterintuitive behavior demonstrates that one cannot predict the degree of ionization solely based on the cross-section of the neutral atom, as often done in an approximate fashion, and that ab initio calculations are needed to quantify the degree of ionization.

In summary, we report experimental and theoretical results on the multiphoton inner-shell ionization of a high-$Z$ atom, xenon, at experimental parameters relevant for molecular structure determination via femtosecond coherent diffractive imaging and for other high-fluence, hard X-ray experiments. The peak photon fluence is two orders of magnitude higher than in previously reported experiments, which leads to the observation of resonance-enhanced X-ray multiple ionization in the hard X-ray regime. The enhancement is highest for photon energies several hundred electron-volts above the $L$-shell ionization threshold. It becomes smaller at higher photon energies as changes in the electronic structure narrow down the range of charge states with resonant transitions and move the resonances to higher, less-accessible charge states. The good agreement of the experimentally observed CSDs with our calculations, in particular for three pronounced peaks at high charge states at 5.5 keV photon energy, underscores the importance of resonant and relativistic effects on

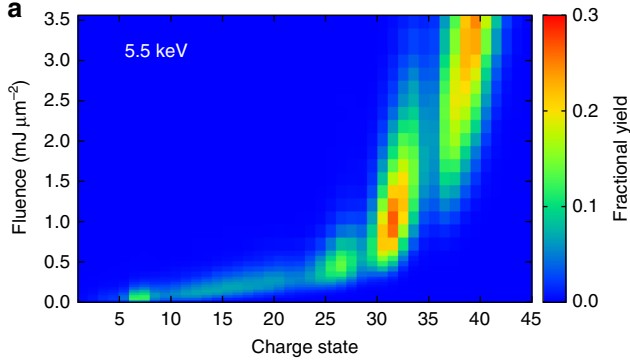

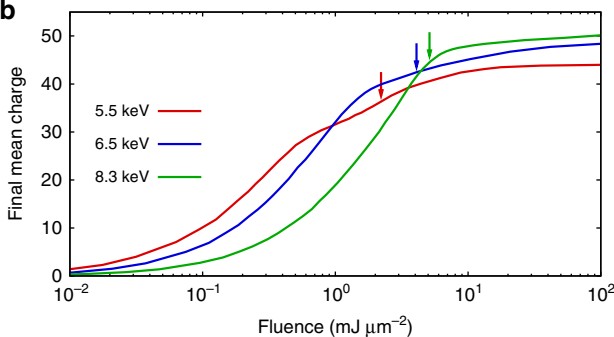

**Fig. 4** Fluence dependence of the multiple ionization. **a** Calculated Xe charge state distribution at 5.5 keV as a function of photon fluence; **b** calculated final mean charges of Xe at three photon energies. Calculations are done with both resonance and relativistic effects. The pulse duration is 30 fs (FWHM), and no focal volume averaging is applied. The arrows indicate the peak fluences used in experiment

the hard X-ray multiphoton ionization of heavy atoms. The strong resonant enhancement for certain charge states allows to map the electronic structure of extremely highly charged ions. Furthermore, our results demonstrate the interplay of intensity and photon-energy dependence for X-ray ionization of heavy atoms and allow to predict, quantify, and possibly minimize the radiation damage, for example in high-fluence, hard X-ray imaging experiments.

## Methods

**Experiment**. The experiment was performed at the Coherent X-ray Imaging (CXI) instrument[24,25] of the LCLS free-electron laser facility at SLAC, operated at 120-Hz repetition rate and with photon energies of 5.5, 6.5, 7, 7.5, and 8.3 keV. The photon energy bandwidth was estimated to be 1% (full-width at half-maximum, FWHM). The nominal pulse duration is 30 fs (FWHM). Using the 0.1 micron Kirkpatrick-Baez mirror system, the LCLS X-ray beam was focused in the center of an ion time-of-flight mass spectrometer, which is described in ref. [5] A 1 mm times 15 mm wide slit aligned perpendicular to the X-ray beam direction in the first spectrometer electrode was used to limit the spectrometer acceptance in order to reduce the ion current on the detector and to exclude ions generated out of focus. The CXI vacuum chamber was backfilled with xenon gas at a pressure of $2.3 \times 10^{-6}$ torr. The xenon ions were generated by the pulsed X-ray beam and extracted by a 83 mm long constant and homogeneous electric field of 22.8 V cm$^{-1}$. The ions passed a 315 mm long drift tube and were detected by a time- and position-sensitive 80-mm microchannel plate (MCP) detector equipped with a delay-line anode.

For each FEL shot, the MCP traces were recorded with an Acqiris DC282 digitizer. Each ion hit was identified in the post-analysis using a constant-fraction discriminator implemented in the software suite CASS[26,27]. The discriminator parameters were optimized for multi-hit detection, and the threshold was adjusted as a function of the time-of-flight in order to account for the velocity-dependent MCP detection efficiency. Signatures of seven xenon isotopes were observed. At increasing charge states, i.e., shorter flight times, the peaks of different isotopes of neighboring charge states start to overlap, and a deconvolution based on the natural isotopic abundance was performed to determine the ion yield for each

charge state. The deconvolution for the 5.5-keV spectrum is illustrated in Supplementary Figure 1.

The shot-by-shot X-ray pulse energies were measured by gas detectors located upstream of the beamline optics. To extract the experimental CSDs shown in Figs. 2 and 3, only those shots with pulse energies within the range specified in Table 1 were selected in the analysis. To derive the actual X-ray fluence distribution (i.e., the energy per unit area at each position within the focal volume) in the interaction zone, which needs to take into account transmission losses in the beamline and the focal profile in the interaction region, calibration measurements on argon atoms were performed under the same experimental conditions.

**Focus parametrization**. The XATOM toolkit[20–22] calculates the CSD of atoms ionized by an XFEL beam for given values of the photon fluence. For a comparison to experimental CSDs, the theoretical results must be averaged over the fluence distribution in the focal volume. In this work, the fluence distribution is described by a double Gaussian[28] in the two dimensions perpendicular to the beam propagation direction. The fluence values in the third dimension along the propagation axis are assumed to be constant since the Rayleigh length is larger than or comparable to the 1 mm entrance slit width of the spectrometer. The second Gaussian is introduced to model the low-fluence regions of the CXI nanofocus beam profile[29]. The double Gaussian profile consists of a linear combination of two independent Gaussians, which can be parameterized by three parameters: the peak fluence, the fluence ratio, and the width ratio[30]. The calibration of the focal volume was conducted using the XCALIB toolkit[30]. It optimizes the parameter set by minimizing the squared, nonlinear difference between theoretical and experimental argon CSDs, which were measured for calibration purposes. Calibrating the focal spot through CSDs of light elements is advantageous because their atomic properties are well known, and the CSD directly reflects the experimental conditions at the focal spot. To calibrate the high fluence regime, only highly charged argon ions were included in the calibration procedure (+11 to +18 for 5.5 keV, +10 to +18 for 6.5 keV, and +9 to +17 for 8.3 keV). Assuming a constant 0.35 μm × 0.3 μm focus size (FWHM) for all photon energies, XCALIB found the best fit to the experimental argon CSD with the peak fluence, width ratio, and fluence ratio listed in Table 1 for each photon energy. The beamline transmission is calculated with these fitted parameters[30]. The calibration reveals a decrease in beamline transmission for decreasing photon energy, which is expected due to an increase in the size of the unfocused X-ray beam at low photon energies, leading to an overfilling of the focusing mirrors. However, since no independent measurement of the focus size at each photon energy was performed, we note that the apparent decrease in beamline transmission is, most likely, also partly due to an increase in focus size. Our calibration cannot disentangle both effects since only the fluence distribution can be derived from the calibration.

**Modeling**. The multiple ionization dynamics of Xe atoms under the irradiation of ultra-intense X-rays are calculated using the XATOM toolkit[20–22], which was recently extended to include relativistic effects and resonant excitations[17]. The relativistic energy corrections are calculated within first-order perturbation theory, and the relativistic electronic configurations are considered with spin–orbit splittings. In the course of the sequential multiphoton absorption process, all but the electrons in the $K$-shell of Xe can be ionized for the photon energies considered in this study. The $q$-hole electronic structure of Xe$^{q+}$ is calculated on the basis of the Hartree–Fock–Slater method including the Latter tail correction in order to improve the asymptotic behavior of the effective potential and, thus, the calculated orbital eigenvalues (see ref. [21] for further details). Atomic data (photo-ionization cross-sections, Auger and Coster–Kronig rates, and fluorescence rates) are calculated in the lowest order perturbation theory. The total number of electronic configurations formed by removing 0, 1, or more electrons from Xe subshells ($n \geq 2$) is 23,532,201 in the non-relativistic case without resonances. This number, which is equivalent to the number of coupled rate equations to be solved, increases to 5,023,265,625 when spin–orbit splittings are considered[17]. The number explodes when including resonant excitations in addition to the spin–orbit splittings. The maximum values of the principal quantum number and the orbital angular momentum quantum number employed in our calculations are $n_{max} = 30$ and $l_{max} = 7$ to obtain the converged total photoabsorption cross-sections shown in Fig. 2b. The total number of electronic configurations formed by removing 0, 1 or more electrons from initially occupied subshells of Xe$^{q+}$ and placing them into unoccupied $(n, l)$-subshells is estimated to be ~$2.6 \times 10^{68}$, according to the number partitioning expression used in ref. [16] To handle such a gigantic number of the rate equations, we apply the Monte Carlo on-the-fly approach, where only the probable pathways of ionization dynamics are followed and the atomic data are calculated only when the electronic configuration is visited in the pathway[17,18]. Thus, this approach imposes no limitation on the size of the configurational space. Our current model does not include higher order many body processes such as shake-off and double Auger decay. Collisional processes such as dielectric recombination, impact ionization, and charge exchange are neglected because the mean free path of electrons and ions exceeded the sub-micron focus size by several orders of magnitude.

## Data availability
The datasets generated during and/or analyzed during the current study are available from the corresponding author on reasonable request.

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

## Acknowledgements
This work is supported by the US Department of Energy, Office of Science, Basic Energy Sciences, Chemical Sciences, Geosciences, and Biosciences Division, who support the Kansas group under contract no. DE-FG02-86ER13491 and the Argonne group under contract no. DE-AC02-06CH11357, and by the excellence cluster "The Hamburg Centre for Ultrafast Imaging: Structure, Dynamics, and Control of Matter at the Atomic Scale" of the Deutsche Forschungsgemeinschaft. Use of the Linac Coherent Light Source (LCLS), SLAC National Accelerator Laboratory, is supported by the US Department of Energy, Office of Science, Office of Basic Energy Sciences under contract no. DE-AC02-76SF00515. A.R. acknowledges support from the National Science Foundation EPSCoR Track II Award No. IIA-1430493. T.G. acknowledges the Peter Ewald Fellowship from the Volkswagen foundation. D.R. acknowledges support from the Helmholtz Gemeinschaft through the Helmholtz Young Investigator Program. K.U. acknowledges the X-ray Free Electron Laser Priority Strategy Program of the Ministry of Education, Culture, Sports, Science and Technology of Japan (MEXT), the Research Program of Dynamic Alliance (five star alliance) for Open Innovation Bridging Human, Environment and Materials in Network Joint Research Center for Materials and Devices, and the TAGEN project for support. We are grateful to the SLAC staff for their support and hospitality during the beamtime and to Evgeny Savelyev for help during the experiment.

## Author contributions
B.R., A.R., and D.R. conceived the experiment, which was coordinated by A.R. and D.R., and carried out by B.R., L.F., B.E., R.B., C.Bom., J.C., S.C., S.B., G.J.W., K.R.F., R.A.-M., J.E.K., T.G., M.B., C.S.L., B.K., S.H.S., L.Y., C.Bos., K.U., T.M., M.S., A.R., and D.R. The time-of-flight spectrometer was assembled and operated mainly by B.R., B.E., R.B., C.Bom., A.R., and D.R. The LCLS CXI beamline and related LCLS equipment was mainly operated by S.B., G.J.W., R.A.-M., and J.E.K. Data acquisition was coordinated by L.F. and S.C. B.R. analyzed the data with help from L.F. K.T. conducted the calibration of the pulse parameters with the atomic argon data using XCALIB developed by K.T. and Z.J. and carried out the calculations using the XATOM toolkit developed by S.-K.S., K.T. and R.S. B.R., K.T., A.R., S.-K.S., R.S., and D.R. interpreted the results and wrote the manuscript with input from all authors.
