## [Peer Review File · Nature Communications]

Reviewers' comments:

Reviewer #1 (Remarks to the Author):

The authors report on the experimental and theoretical study of the multiple ionization of single xenon atoms upon exposure to ultra intense and short hard X-rays ranging from 5.5 to 8.3 keV from an XFEL. The comparison of theory and experiment shows, that for the absorption of multiple X-ray photons relativistic and resonant effects need to be taken into account, as predicted in a previous publication by the team. Structured charge state distributions for three different photon energies are observed, which indicate the resonant positions of relativistically shifted electronic levels in the sequence of ionization processes illustrated in the appendix.

The work is carried out by a highly competent team of experts in the field. The well-established experiments and calculations are rigorous and of high quality. A well-written introduction is given, and the list of references is comprehensive. The article is clearly structured as its predecessors and can be readily followed. The spectra and tables are informative and comprehensible for the most part (see the few comments below).

The rather mature experimental and theoretical work can be seen as a nice and solid continuation of related previous projects on krypton and xenon with familiar spectra and sections (publications [12, 13, 18, and 19], as referenced in the draft, explain the concepts on measurements with lower fluence). The draft discusses interesting observations in terms of (hidden) resonances and is valuable for the community. It should thus be published. Does this article qualify for the criteria of publication of Nature Communications? While the study provides strong evidence of its conclusion, I don't think that it is as novel and of extreme importance to scientists in the field beyond the previously gained knowledge, as demanded by Nature Communications. Due to its rather incremental advancement I think, that it won't be interesting to researchers in other related disciplines. When reflecting on the literature (e.g. the aforementioned publications by the team members) I don't see a very strong case, neither in terms of novelty, experimental/theoretical progress or fundamental new physics revealed. The authors state themselves that "...in spite of the complexity of the problem, our current ionization model can fully explain the experimental findings for all X-ray beam parameters." and present the underlying processes as late as figure 6 in the Supplemental Material section, since they were mostly covered by the original REXMI mechanism discussed in [12]. This is nice work and a nice article, but I am hence inclined to recommend publication in a more specialized journal.

By all means, the draft is well written like all the related previous publications; I have only very few cosmetic suggestions:

- At the end of the first paragraph on page 5 I suggest to say "...stand out from this steady decrease as seen in figure 2a."
- The legends in figure 2a and 2c are hard to read. Either create a bigger legend to the side or implement this information in the figure caption. I find the color purple too similar to the red color of the competing line. Perhaps the color grey would be a better choice.
- In the second paragraph on page 6 I recommend to write "...and relativistic effects (see color coding in the legend) are presented in comparison with the experimental CSD (see black)."
- In the figure caption of figure 6 please either list the six X-ray absorption processes considered here or refer to the text for an explanation of the abbreviations.

Reviewer #2 (Remarks to the Author):

The authors have performed multiple ionization studies of xenon with x-ray photons spanning 5.5 to 8.3 keV at very high fluence values (some mJ per square micron). In this combination this is a new experiment. However, the importance of bound-bound resonances in highly charged ions in multiple keV photon absorption has already been shown by the authors (Refs 12 and 13) while the 5 keV barrier has already been overcome for multiphoton ionization in Refs 19 and 20. As in earlier works, the measured ionic charge state distributions (CSDs) have been compared to calculations from the authors' XATOM toolkit. Compared to the earlier works cited above, in the present manuscript relativistic effects on ionic levels were included which largely improves the agreement between experiment and theory, especially for the highest possible charges and for the position of peaks in the CSDs. However, the relativistic extension has already been published (Ref 18). Therefore, many findings of the manuscript are rather incremental. New is the (theoretic) study of the fluence dependence of the CSDs, which has a surprising result that is physically well founded. It is a pity that these calculated fluence-dependent CSDs are so different from experimental one because of the focal averaging. Maybe it would help to include one calculated CSD without focal averaging to one of the experimental figures so that readers can get an impression of focal averaging effects on the distributions. In the supplementary material, an experimental fluence dependence is reported which is certainly interesting to the field. However, it does not seem to be mentioned in the main text.

Overall, the paper is very well written and presents sound data in a convincing way. I am, however, a bit concerned about the claims on statistical errors below 1% for all charge state fractions. The authors do not support their claim by a statistical analysis of the data and especially their delicate deconvolution technique to disentangle the overlapping ToF-spectra at very high charge states. No analysis for the statistical errors of the simulations are given, either. On the other hand, I am wondering how the authors explain the apparent deviations between experiment and the (most sophisticated) model that are well beyond the 1% statistical errors, especially at lower charge states. Are there systematic errors? Please discuss this.

Apart from that I have some minor comments:

- In some figures the key is so small that it is hardly readable.
- On page 6, when theoretical results are first introduced, please mention focal averaging
- On page 7 second paragraph, a cutoff of charge state 22+ without resonances is claimed, but figure 2a suggests 35+
- On page 7 third paragraph mention that the $n=4$ levels are orange in figure 2c.
- On page 7 third paragraph: Why are resonances from the L shell to shells with n larger than 4 not discussed? Do they not enhance ionization? As seen in fig 2c, they should come into resonance at lower charge states.
- On page 10 in the summary a sentence reads "The enhancement is highest for photon energies several hundred eVs above the inner-shell ionization threshold" Does this refer to any inner-shell ionization threshold or to a specific one? Please clarify.
- Not all fluence values for all photon energies are given.

Reviewer #3 (Remarks to the Author):

The manuscript reports a combined experimental-theoretical study on the multiple ionization of xenon atoms by intense femtosecond X-ray pulses. Emphasis is put on the importance of resonant and relativistic effects, both of which are found to be indispensable for modeling the experimental data.

The manuscript is well structured and written. The work is of high relevance to theory and experiment in physics and chemistry and most of the results are presented and discussed in an adequate manner. However, I suggest the authors work on the following issues before the manuscript is considered further for publication:

- Figure 2 presents experimental and computed charge state distributions. On p. 7/8 the authors claim that the peaks in the charge state distribution are connected to resonant transitions in the species that have one charge less. This is convincing for $\text{Xe}^{30+}/\text{Xe}^{31+}$ and $\text{Xe}^{36+}/\text{Xe}^{37+}$. However, the hypothesis does not work well for the third peak: The maximum in the cross section (Fig. 2b) is calculated to be at Xe^{27+} but the maximum in the charge state distribution (Fig. 2a) occurs at Xe^{26+} and not at Xe^{28+} . This inconsistency should be pointed out in the manuscript. Do the authors have an explanation?

- The authors put a lot of emphasis on the importance of relativistic effects but it is in fact never stated in what way they are considered in the theoretical treatment. Ref. 18 probably contains that information, but one or two sentences about the treatment of relativistic effects ought to be included in the modeling section of the present manuscript.

- Along the same lines: Figure 2 is supposed to illustrate the importance of relativistic effects for modeling the charge state distribution. I think this claim would be stronger if the authors explained what transitions cause the peaks in the non-relativistic treatment (pink curve). This curve seems to provide already a good zeroth-order description. Is it qualitatively wrong nevertheless?

- The electronic structure of Xe^{q+} is calculated on the basis of the Hartree-Fock-Slater method (p.14). This is considered inadequate and outdated in electronic structure theory. The work would benefit from using modern methods of quantum chemistry.

- The statements on p.15 "this approach imposes no limitation on the size of the configurational space" and "Our current model does not include higher order many body processes such as shake-off and double Auger decay" contradict each other. Neglect of higher order many body processes such as shake-off states and double Auger decay does constitute a restriction of the configurational space.

- In the section about resonance-induced processes (supplementary material, p.19), the authors state that "For most cases, A' (spectator Auger decay) is the dominant process". Table II shows that A' is in fact the dominant process in two out of three cases. Also, the dominance is not very pronounced and there are no clear trends in Table II. Therefore, I think the statement on p.19 is too strong. The authors should present further evidence for the dominance of A' or modify the discussion appropriately.

Reviewer #4 (Remarks to the Author):

The paper under consideration reports on a combined experimental/theoretical study of multiple ionization of xenon atoms by ultra-intense femtosecond FEL pulses in the hard x-ray regime. Higher peak fluences than in previous experiments have been achieved which provides access to a hitherto unexplored regime involving extremely high ionization multiplicities, i.e., xenon ion charge states up to 50. The experimental charge-state distributions show rich structure and the comparison with state-of-the-art theory demonstrates that interesting physics is responsible for this. The authors have carried out four types of calculations, excluding and including relativistic and resonance effects. Only the most advanced calculation that includes both gives good agreement with the measurements. This,

together with the subsequent analysis, provides conclusive evidence for the crucial role of these effects in the creation of high charge states.

This is an exciting (and somewhat frightening) result demonstrating the complexity of multiple ionization of heavy atoms in ultra-strong, hard x-ray fields. The authors do an excellent job in analyzing and explaining their results (the analysis is deep, yet presented in a concise way) and there is no doubt in this referee's mind that the work is sound and of interest to the community.

There is in fact only one point that I would like the authors to consider and address in a reply to this report, or better, in a revised version of their paper: Their 'best' calculation (red curves in Figs. 2 and 3) gives a good description of the experimental distributions at high charge states (that's where the relativistic and resonance effects are at play). But around the main (single-photon) peak the agreement is not so good, in particular the theory (all four models coincide here) misses the position of the experimental peak at charge state $q=8$. This is not discussed (except for a comment on Xe(8+) on p.23 that may be related to this). I think some mention of this issue in the main part of the paper would be appropriate.

Other than that I have one question (a) and four minor points (b-e):

(a) In the Methods section it is stated that the Hartree-Fock-Slater (HFS) method was used to calculate the q -hole structure of the ions. Slater exchange gives rise to an incorrect asymptotic behavior of the effective potential and this affects the accuracy of orbital eigenvalues. There are ways to correct for this and it would be good to know what was done in this work.

(b) I don't think "wildly" (p.5, second line) is an appropriate characterization of what is seen in Fig. 1.

(c) Replace "charged state" by "charge state" in the caption of Table II.

(d) At all places except one (last line of p. 22) the photon fluence is given in units of $\text{mJ}/\mu\text{m}^2$. Is this single departure from this rule necessary?

(e) I understand that the slopes given in the legend of Fig. 8 correspond to the number of absorbed photons. But if I don't read the text and just look at the figure I wouldn't expect the slopes to be dimensionless. Can this be explained?

Reviewer #1:

At the end of the first paragraph on page 5 I suggest to say "...stand out from this steady decrease as seen in figure 2a."

We have added the reference to Fig. 2a as suggested.

The legends in figure 2a and 2c are hard to read. Either create a bigger legend to the side or implement this information in the figure caption. I find the color purple too similar to the red color of the competing line. Perhaps the color grey would be a better choice.

We have increased the font size of the legend and have changed the purple line to a grey line in figure 2a, 2b, and 3.

In the second paragraph on page 6 I recommend to write "...and relativistic effects (see color coding in the legend) are presented in comparison with the experimental CSD (see black)."

We have changed the text accordingly.

In the figure caption of figure 6 please either list the six X-ray absorption processes considered here or refer to the text for an explanation of the abbreviations.

We have added an explicit explanation of all six processes in the caption of figure 6.

Reviewer #2:

Maybe it would help to include one calculated CSD without focal averaging to one of the experimental figures so that readers can get an impression how focal averaging affects the distributions.

The calculated, fluence-dependent CSDs at 5.5 keV photon energy without focal averaging are shown in Figure 4a. At a single fluence, the CSDs are dominated by only a few charge states. When including focal averaging, we obtain the CSD shown in Figure 2a, which spans a large range of charge states and which is in good agreement with the experimental CSD.

In the supplementary material, an experimental fluence dependence is reported which is certainly interesting to the field. However, it does not seem to be mentioned in the main text.

We have added a reference to this figure in the main text in the context of discussing Figure 4a.

I am, however, a bit concerned about the claims on statistical errors below 1% for all charge state fractions. The authors do not support their claim by a statistical analysis of the data and especially their delicate deconvolution technique to disentangle the overlapping ToF-spectra at very high charge states. No analysis for the statistical errors of the simulations are

given, either. On the other hand, I am wondering how the authors explain the apparent deviations between experiment and the (most sophisticated) model that are well beyond the 1% statistical errors, especially at lower charge states. Are there systematic errors? Please discuss this.

The time-of flight spectra were integrated within the indicated pulse energy interval and then divided by the number of shots within this interval. Figure 1 shows such a normalized ion yield. For the data shown in figure 2a, we measured 29100 shots within the chosen pulse energy interval and counted, for example, 7275 Xe⁴⁰⁺ ions, corresponding to an ion yield of 0.25 Xe⁴⁰⁺ ions per shot. The statistical uncertainty is, thus, $\sqrt{7275}=85$, which is indeed slightly larger than 1% in this case. The highest charge state displayed in figure 2a is 42+, which has a statistical uncertainty of 3%. All other error bars are below 1%. The caption of figure 2a has been changed accordingly. Regarding the deviations between experiment and calculations at lower charge states, these stem from the fact that higher-order many-body processes, such as shake-off and double Auger decay, and other higher-order many-electron corrections are not included in our calculations. As we have noted in previous applications of the XATOM code, this often leads to deviations in the low-charge-state region, which is dominated by single-photon ionization. We have included a sentence in the main text to clarify this: “We note that in the low-charge-state region below Xe¹³⁺, which is dominated by single-photon ionization, all four calculations show deviations from the experimental CSD as a result of neglecting higher-order many-electron corrections. Such effects are expected to lose relevance for higher charge states, where fewer and fewer electrons remain bound.”

In some figures the key is so small that it is hardly readable.

We have increased the font size (12 points) in all figures.

On page 6, when theoretical results are first introduced, please mention focal averaging.

We have added a sentence mentioning the focal averaging in the paragraph where the theoretical results are introduced.

On page 7 second paragraph, a cutoff of charge state 22+ without resonances is claimed, but figure 2a suggests 35+.

For the relativistic case, +22 is the charge state at which direct single photon ionization from the L-shell terminates (thus the term “L-shell ionization cutoff”). The cross section drops significantly at this charge state, as shown in figure 2b. However, even when excluding resonant excitations, charge states higher than +22 can be produced by multiple-core-hole formation and valence photoionization, with the latter being the dominant process as long as valence photoionization is possible. Since the wording in the previous version was indeed misleading, we have deleted the phrase referring to the “L-shell ionization cutoff at Xe²²⁺”, which removes the ambiguity.

On page 7 third paragraph mention that the n=4 levels are orange in figure 2c.

We have added this in parentheses.

On page 7 third paragraph: Why are resonances from the L shell to shells with n larger than 4 not discussed? Do they not enhance ionization? As seen in fig 2c, they should come into resonance at lower charge states.

Supplementary figure 7 shows all contributions of transitions with higher n ($4 \leq n \leq 7$) to the calculated resonant excitation cross section. As the referee pointed out, resonances from the L-shell to $n > 4$ start at lower charge states. For example, there is a small hump around $+9 \sim +13$ in the calculated cross section due to the resonance from $n=2$ to $n=5$ (red). In addition, there are more transitions from the L-shell to $n > 4$ for $+16 \sim +30$, especially the transitions from $n=2$ to $n=5$ (red) around $+24$, which contributes to the first peak in the CSD at Xe^{26+} . To clarify this point, we have modified the description of the three maxima as follows: “As a consequence of the increased cross section, the CSD also exhibits three maxima centered at the charge states directly following these resonance positions, except for the first maximum around Xe^{26+} , where several transitions from the L-shell to $n > 4$ also contribute at slightly lower charge states.”

On page 10 in the summary a sentence reads "The enhancement is highest for photon energies several hundred eVs above the inner-shell ionization threshold" Does this refer to any inner-shell ionization threshold or to a specific one? Please clarify.

We have added the word “L-shell” to clarify which threshold we are referring to in this particular case. However, based on the present results (Xe L-shell at 5.5 keV) and our findings in Ref. [12] (Xe M-shell at 1.5 keV) and Ref. [13] (Kr L-shell at 2 keV), we believe that the statement is generally valid for any inner-shell ionization threshold.

Not all fluence values for all photon energies are given.

Since we did not take calibration data at 7 eV and 7.5 keV, the fluence values for these two photon energies are not shown in the table I. We have clarified this in the caption of the table I.

Reviewer #3:

Figure 2 presents experimental and computed charge state distributions. On p. 7/8 the authors claim that the peaks in the charge state distribution are connected to resonant transitions in the species that have one charge less. This is convincing for $\text{Xe}^{30+}/\text{Xe}^{31+}$ and $\text{Xe}^{36+}/\text{Xe}^{37+}$. However, the hypothesis does not work well for the third peak: The maximum in the cross section (Fig. 2b) is calculated to be at Xe^{27+} but the maximum in the charge state distribution (Fig. 2a) occurs at Xe^{26+} and not at Xe^{28+} . This inconsistency should be pointed out in the manuscript. Do the authors have an explanation?

As already mentioned above in one of our answers to Reviewer #2, resonant excitations to $n > 4$ levels, which occur around a charge state of $+24$, also contribute to the formation of the first hump in the CSD. To make this point clearer, we have modified the manuscript as follows: “As a consequence of the increased cross section, the CSD also exhibits three maxima centered at the charge states directly following these resonance positions, except for the first maximum around

Xe²⁶⁺, where several transitions from the L-shell to n>4 also contribute at slightly lower charge states.”

The authors put a lot of emphasis on the importance of relativistic effects but it is in fact never stated in what way they are considered in the theoretical treatment. Ref. 18 probably contains that information, but one or two sentences about the treatment of relativistic effects ought to be included in the modeling section of the present manuscript.

We have added a short description of our relativistic treatment in the Modeling section of the Methods: “The relativistic energy corrections are calculated within first-order perturbation theory, and the relativistic electronic configurations are considered with spin-orbit splittings.”

Along the same lines: Figure 2 is supposed to illustrate the importance of relativistic effects for modeling the charge state distribution. I think this claim would be stronger if the authors explained what transitions cause the peaks in the non-relativistic treatment (pink curve). This curve seems to provide already a good zeroth-order description. Is it qualitatively wrong nevertheless?

The results of the non-relativistic, resonant calculations is qualitatively wrong because it shows four peaks in the structured CSD instead of three. We have clarified this in the text and added a brief explanation of the non-relativistic transitions corresponding to each peak: “The non-relativistic, resonant calculation (grey) produces four peaks (mainly corresponding to 2s→5p, 2p→5d, 2s→4p, and 2p→4d transitions) instead of three and, thus, does not reproduce the experimental data.” (Note that the pink color of the non-relativistic, resonant calculation has been changed to a grey color in response to the request by reviewer #1.)

The electronic structure of Xe^{q+} is calculated on the basis of the Hartree-Fock-Slater method (p.14). This is considered inadequate and outdated in electronic structure theory. The work would benefit from using modern methods of quantum chemistry.

Modern quantum chemistry methods like CI and MCSCF are suitable for calculating the ground state and the first few excited states, whereas we have to support highly excited multiple-hole states produced by the interaction with ultra-intense X-rays. Furthermore, we need to consider a tremendous number of excited states during the X-ray multiphoton ionization dynamics. The computational effort of calculating all of them with quantum chemistry codes is currently prohibitive. Although the Hartree-Fock-Slater method is certainly less accurate (even when including the Latter tail correction, as done in our calculations – see also our response to Reviewer #4), our experience from this work as well as from our past work has shown that the inaccuracies of this method generally play only a minor role when studying X-ray multiphoton ionization dynamics that involve transitions of several hundred eV to a few keV and a relatively large photon bandwidth.

The statements on p.15 "this approach imposes no limitation on the size of the configurational space" and "Our current model does not include higher order many body processes such as shake-off and double Auger decay" contradict each other. Neglect of

higher order many body processes such as shake-off states and double Auger decay does constitute a restriction of the configurational space.

Our Monte Carlo on-the-fly approach can, in principle, handle all possible electronic configurations that are formed by putting 0, 1 or more electrons into an Xe subshell. This includes not only those subshells that are initially occupied, but also subshells that are initially unoccupied. In this sense, there is no limitation on the size of the configurational space. We do not include higher order many body processes, which makes a restriction on the treated x-ray-induced processes, but not on the configurational space.

In the section about resonance-induced processes (supplementary material, p.19), the authors state that "For most cases, A' (spectator Auger decay) is the dominant process". Table II shows that A' is in fact the dominant process in two out of three cases. Also, the dominance is not very pronounced and there are no clear trends in Table II. Therefore, I think the statement on p.19 is too strong. The authors should present further evidence for the dominance of A' or modify the discussion appropriately.

The statement has been modified and now reads: "In all three cases, A' (spectator Auger decay) is the dominant process leading to further ionization."

Reviewer #4:

There is in fact only one point that I would like the authors to consider and address in a reply to this report, or better, in a revised version of their paper: Their 'best' calculation (red curves in Figs. 2 and 3) gives a good description of the experimental distributions at high charge states (that's where the relativistic and resonance effects are at play). But around the main (single-photon) peak the agreement is not so good, in particular the theory (all four models coincide here) misses the position of the experimental peak at charge state $q=8$. This is not discussed (except for a comment on Xe(8+) on p.23 that may be related to this). I think some mention of this issue in the main part of the paper would be appropriate.

Neglecting higher-order many-body processes (such as shake-off and double Auger decay, for instance) and other higher-order many-electron corrections, which are not included in our calculations, can lead to deviations in the low-charge-state region, which is dominated by single-photon ionization. In the present case, it leads to a maximum of the calculated CSD at $q=7$ instead of $q=8$ and to an underestimation of the yield of the following charge states up to $q=12$, as was also seen in Ref. [19], for example. As already mentioned above in our response to Reviewer #2, who had a similar question, we have included a sentence in the main text to clarify this: "We note that in the low-charge-state region below Xe¹³⁺, which is dominated by single-photon ionization, all four calculations show deviations from the experimental CSD as a result of neglecting higher-order many-electron corrections. Such effects are expected to lose relevance for higher charge states, where fewer and fewer electrons remain bound."

(a) In the Methods section it is stated that the Hartree-Fock-Slater (HFS) method was used to calculate the q-hole structure of the ions. Slater exchange gives rise to an incorrect asymptotic behavior of the effective potential and this affects the accuracy of orbital eigenvalues. There are ways to correct for this and it would be good to know what was done in this work.

To correct the asymptotic behavior, we used the Latter tail correction, which was specified in our original implementation of XATOM [22]. We have added a brief statement about this in the Methods: “The q-hole electronic structure of Xe^{9+} is calculated on the basis of the Hartree-Fock-Slater method including the Latter tail correction in order to improve the asymptotic behavior of the effective potential and, thus, the calculated orbital eigenvalues (see Ref. [22] for further details).”

(b) I don't think "wildly" (p.5, second line) is an appropriate characterization of what is seen in Fig. 1.

We have replaced the word “wildly” by “non-monotonically”.

(c) Replace "charged state" by "charge state" in the caption of Table II.

This mistake has been corrected.

(d) At all places except one (last line of p. 22) the photon fluence is given in units of mJ/micron². Is this single departure from this rule necessary?

Since the photon fluence to saturate single-photon absorption is simply given by the inverse of the photoabsorption cross section [5.38×10^{10} (photons/ μm^2) = $1/0.186$ (Mb) $\times 10^{10}$], we felt that it was more intuitive to state both, the photon fluence (photon/ μm^2) and the energy fluence ($\mu\text{J}/\mu\text{m}^2$), in this case.

(e) I understand that the slopes given in the legend of Fig. 8 correspond to the number of absorbed photons. But if I don't read the text and just look at the figure I wouldn't expect the slopes to be dimensionless. Can this be explained?

Figure 8 is a log-log plot. If a function $y(x)$ is plotted on a log-log scale, what is meant by the slope at position x is $\log[y(x+dx)/y(x)] / \log[(x+dx)/x]$ for sufficiently small dx . Because the slope is a function of dimensionless ratios, it is dimensionless. In a log-log plot, a power law of the form $y(x) = a \cdot x^b$ is represented by a straight line. The dimensionless exponent b is the slope of that straight line.

REVIEWERS' COMMENTS:

Reviewer #2 (Remarks to the Author):

From my point of view, the authors have adequately addressed all points of concern, putting the paper in a very solid state. However, I still consider their main results rather incremental but surely with relevance for high-intensity X-ray applications.

Reviewer #3 (Remarks to the Author):

I went over the response letter and the revised manuscript. In my view, the authors have done an adequate job in addressing the points raised by me and the other referees. I believe the manuscript now meets the criteria for publication in Nature Communications; I recommend acceptance.

Reviewer #4 (Remarks to the Author):

I have read the revised manuscript and the detailed response letter of the authors which addresses all points and questions raised by the referees. In the first part of their letter the authors provide good reasons for why their work is suitable for publication in Nature Communications. I find their case quite convincing.

I am also happy with their point-by-point responses. As far as I can see they have addressed and dealt with all points raised in a satisfactory manner (certainly with mine). In my view the paper can now be published as is.